# Comparative Sensing and Judgment Control System for Temperature Maintenance for Optimal Treatment in Hyperthermic Intraperitoneal Chemotherapy Surgery

**DOI:** 10.3390/s24020596

**Published:** 2024-01-17

**Authors:** Tae-Hyeon Lee, Kicheol Yoon, Sangyun Lee, Woong Rak Choi, Kwang Gi Kim

**Affiliations:** 1Department of Electronic Engineering, Gyeonggi University of Science and Technology, 269 Gyeonggigwagi–dearo, Gyeonggi–do, Siheung City 15073, Republic of Korea; thlee@gtec.ac.kr; 2Medical Devices R&D Center, Gachon University Gil Medical Center, 21, 774 beon–gil, Namdong–daero Namdong–gu, Incheon 21565, Republic of Korea; kcyoon98@gachon.ac.kr (K.Y.); l0421h@gmail.com (S.L.); woong97529@korea.ac.kr (W.R.C.); 3Department of Premedicine Course, College of Medicine, Gachon University, 38–13, 3 Dokjom–ro, Namdong–gu, Incheon 21565, Republic of Korea; 4Department of Health and Safety Convergence Sciences & Health and Environmental Convergence Sciences, Korea University, 145, Anam-ro, Seongbuk-gu, Seoul 02841, Republic of Korea; 5Department of Biomedical Engineering, Gachon University, 38–13, 3 Dokjom–ro, Namdong–gu, Incheon 21565, Republic of Korea; 6School of Electrical Engineering, Korea University, 145, Anam-ro, Seongbuk-gu, Seoul 02841, Republic of Korea; 7Department of Health Sciences and Technology, Gachon Advanced Institute for Health Sciences and Technology (GAIHST), Gachon University, 38–13, 3 Dokjom–ro, Namdong–gu, Incheon 21565, Republic of Korea

**Keywords:** comparator, look-up table, automatic temperature control, thermal chemotherapy, cancer surgery

## Abstract

For tumors wherein cancer cells remain in the tissue after colorectal cancer surgery, a hyperthermic anticancer agent is injected into the abdominal cavity to necrotize the remaining cancer cells with heat using a hyperthermic intraperitoneal chemotherapy system. However, during circulation, the processing temperature is out of range and the processing result is deteriorated. This paper proposes a look-up table (LUT) module design method that can stably maintain the processing temperature range during circulation via feedback. If the temperature decreases or increases, the LUT transmits a command signal to the heat exchanger to reduce or increase heat input, thereby maintaining the treatment temperature range. The command signal for increasing and decreasing heat input is *T_p_* and *T_a_*, respectively. The command signal for the treatment temperature range is *T_s_*. If drug temperatures below 41 and above 43 °C are input to the LUT, it sends a *T_p_* or *T_a_* signal to the heat exchanger to increase or decrease the input heat, respectively. If the drug’s temperature is 41–43 °C, the LUT generates a *T_s_* signal and proceeds with the treatment. The proposed system can automatically control drug temperature using temperature feedback to ensure rapid, accurate, and safe treatment.

## 1. Introduction

The biggest problem after surgery to remove colon cancer is that tiny pieces of residual cancer cells remain attached to surrounding tissues, so the probability of recurrence (within 5 years) is estimated to be more than 50% high [1]. These tiny residual cancer cell fragments are difficult to excise through surgery [1].

Chemotherapy is used simultaneously to reduce the possibility of recurrence, but patients face difficulties during the treatment process due to side effects [1]. In order to pursue methods to promote the inhibition of cancer cell growth, treatment methods currently used in medical institutions include HIFU (high-intensity frequency ultrasound) treatment, laser treatment, and high-frequency treatment [2,3,4,5].

High-frequency heat therapy uses electromagnetic waves to inhibit cancer cell growth. In particular, electromagnetic waves can be irradiated to cancerous tissue to induce heat generation and increase the body’s metabolic rate, thereby inhibiting cancer cell proliferation, leading to an increase in the 5-year survival rate and an improvement in surgical treatment outcomes [6]. Therefore, high-frequency treatment has no side effects such as nausea, vomiting, loss of appetite, weight loss, digestive problems, or hair loss. Nevertheless, red spots may appear on the skin tissue where the electrodes are attached, and side effects such as mild burns, scars, inflammation, and lumps may appear in the fat layer [6,7].

HIFU can suppress cancer cell growth by promoting tumor heat through targeted treatment [8]. These methods promote damage to cancer cells but have the side effect of damaging surrounding skin or normal tissue due to heat [8]. Laser treatment causes a chemical reaction between light and oxygen. This chemical reaction generates instantaneous heat, damaging the microvessels around cancer cells with heat [9]. Therefore, there is a method to minimize the patient’s pain and block the supply of nutrients to the cancer tissue, but it can cause side effects such as skin hypersensitivity, swelling, pain, and pigmentation [10]. In particular, laser treatment may damage the skin [9,10].

To relieve these side effects, hyperthermic intraperitoneal chemotherapy (HIPEC) surgery can be used. The advantage of HIPEC surgery is that it removes malignant tumors through surgery and injects high-temperature anticancer drugs to damage fine cancer pieces with heat, allowing treatment to occur simultaneously during the cancer removal surgery [8,11,12,13,14,15,16]. At this time, high-temperature (46 °C) anticancer drugs are injected into the abdominal cavity through a catheter. The injected high-temperature anticancer agent suppresses the growth of cancer cells by heating the remaining tumor at 41 °C to 43 °C while filling the abdomen. The promoting reason for growth inhibition is that HSP90 growth is inhibited between 41 °C and 43 °C [17,18].

The high-temperature anticancer drug is drained again through the catheter, and the drained anticancer drug is a treatment system that repeats injection and drainage for 90 to 120 min through filtering circulation [11,12,13,14,15,16]. Of course, if the temperature of the anticancer drugs is higher than 46 °C, normal tissues may be damaged, so the optimal temperature range for treatment is defined as 41 °C to 43 °C [11,12,13]. However, if drug inflow and outflow are repeated, the temperature may decrease or increase. Therefore, if the temperature falls outside the appropriate temperature range for treatment (41 °C to 43 °C), treatment may not occur or surrounding tissue may be damaged [19]. Commercial HIPEC devices such as ThermoChem HT–2000 (ThermaSolutions, St Paul, MN, USA), Cavi-therm (Caviterm, Provaglio d’Iseo, Italy), SUNCHIP (Camida Tech., Birmingham, UK), and Belmont Hyperthermia Pump (Belmont Medical, Belmont, MA, USA) are currently in use in operating rooms as systems. The operating status can be monitored, and the temperature must be observed through a sensor and physically adjusted to maintain the appropriate temperature for treatment. However, it is not easy to automatically adjust the optimal treatment temperature for intra-abdominal temperature distribution [20]. In particular, automatic sensors that can detect real-time temperature, devices for comparison and judgment to maintain the appropriate temperature range for treatment (41 °C to 43 °C), and automatic temperature control devices are lacking. It takes a long time to adjust and maintain the appropriate treatment temperature range (41 °C to 43 °C), and there are limits to accurate adjustment, so the accuracy of temperature errors may be reduced. Therefore, to maintain the appropriate temperature for treatment, staff must visually monitor the temperature display attached to the system. The downside is that staff may feel uncomfortable handling the system if they manually control the temperature.

To meet these needs, various HIPEC systems are being developed [8]. Spratt has the ability to maintain temperature regulation and maintenance function through an intraperitoneal heating device or annular phased array technology [8,9,10,11,12,13,14,15,16,19,20,21,22,23]. The purpose of this system is to damage cancer cells with heat by controlling fluid flow to ensure uniform distribution of high-temperature drugs in the abdominal cavity. However, during the treatment process, there is the inconvenience of continuously checking the temperature of the drug with the naked eye, real-time visual monitoring using a temperature sensor to detect the temperature in real-time, and physically adjusting the temperature to maintain the temperature at an appropriate level for treatment. Therefore, it is necessary to develop a system equipped with the performance of automatic temperature control that can automatically detect the drug temperature by applying an automatic control system and maintain the temperature of the injected drug from 41 °C to 43 °C.

This paper proposes the design of a comparator and look-up table (LUT) for real-time temperature detection, comparison/judgment, and temperature control to maintain the optimal temperature range for treatment (41 °C to 43 °C). The proposed system reduces (<41 °C)/increases (>43 °C)/maintains the temperature (design the performance to control the optimal temperature (41 °C to 43 °C) by receiving a command signal from the LUT on whether 41 °C to 43 °C is necessary. Therefore, if the temperature is decreased (<41 °C), it will heat away from the LUT, and if the temperature is increased, the increased temperature will take away the heat from the LUT. If there is no temperature difference, the performance of maintaining the current temperature state (41 °C to 43 °C) can be designed with a comparator and LUT.

## 2. Analysis and Design Method

As shown in Figure 1, the existing system can monitor flow rate and temperature changes for the drug and control temperature. However, manual control was used to change the temperature. However, the proposed system can generate a control signal by comparing and judging the temperature through real-time temperature detection and automatically adjust the temperature by the control signal. Therefore, it is a system that can be adjusted to the appropriate temperature level for treatment.

The proposed HIPEC system consists of a fluid box, heat exchanger, and filter, as illustrated in Figure 2. The inflow and outflow catheters are connected from the system to the abdomen [19]. The fluid box injects the anti-cancer drugs, which are introduced into the heat exchanger through the catheter line (inflow part), as illustrated in the figure [19].

The HIPEC system is intended to maintain drug temperature between 41 and 43 °C in the abdominal cavity and includes a reference heat generator (HG) that generates the desired reference temperature (*T_ref_*) of 41–43 °C, as illustrated in Figure 3. Additionally, it has a temperature-to-digital signal converter (TDC) that converts the data into a digital signal. The outflow temperature (*T_o_*) of the drug from the outside of the abdominal cavity is converted into digital signals (0, 1) by the TDC and introduced into the temperature comparator (TC). The TC compares the *T_ref_* and *T_o_* and outputs pulse signals (0 and 1) indicating temperature difference (*T_D_*).

The *T_o_* signal converted into a pulse signal is input to the TC that consequently generates a *T_D_* signal, as shown in Equation (2), by comparing it with *T_ref_*, as shown in Equation (1) [24,25].
(1)$$T_{D}\left(t\right)=T_{o}squ\left[T_{ref}\left(t\right)+h\left(t\right)\right]$$
(2)$$T_{D}\left(t\right)={\chi T}_{ref}h\left(t\right)-\chi T_{o}\left(t\right)=T_{\chi }T_{ref}h\int_{0}^{t}T_{o}dt$$
where *h* and *χ* represent Joule’s heat and thermal conductivities with respect to heat and *T_o_*, respectively. The output of Equation (1) changes over time through a limited temperature range due to *T_ref_* (t). However, *T_ref_* varies within the desired temperature range. Hence, the temperature range must have excellent thermal conductivity (*χ*) to accurately maintain the desired drug temperature. Further, *h* is defined as the Joule heat function for maintaining the desired temperature range. *T_χ_* is the pulse signal of one period for temperature and is a function of a control signal. It changes the on/off of the pulse according to temperature. Further, squ signifies the function of a signal (pulse) in the form of a square. *T_o_* varies the temperature to rise, fall, and maintain over time. Therefore, the sum of *T_ref_* and *T_D_* (*T_ref_* + *T_D_*) can observe the *T_o_* temporal state of change.

*T_D_* is a result of the difference between *T_ref_* and *T_o_*. Simultaneously, variations in *T_ref_* and *T_o_* over time in the *h* (t) function range with excellent thermal conductivity (*χ*) generate a temperature difference in *T_C_*. *T_D_* (t) varies from 0 to *t*, as given by Equation (2).

As illustrated in Figure 3 and Table 1, *T_D_* is input to the LUT, which outputs the command signal (ord), as specified in Table 1 (refer to the Appendix A on the website). Command signals include the *T_p_* (if drug temperature < 41 °C) to increase heat (1), *T_s_* (if drug temperature is between 41 and 43 °C) to maintain heat (1), *T_a_* (if drug temperature > 43 °C) to decrease heat (0), and *T_x_* (if drug temperature > 44 °C) to stop the system (1).

This command signal is transferred to the heat exchanger, which controls the heat input. Figure 4 illustrates the flow chart of system operation, and Figure 5a,b illustrate the block diagram and schematic of the HIPEC system connected to the TC and LUT. It comprises an RS flip-flop (RS–F/f). Figure 5c shows the simulation results for a logic circuit of Figure 5b. From the figure, the pulse wave can be obtained by the TP_1_, TP_2_, TP_3_, and TP_4_, respectively. When a clock occurs at TP_1_, TP_2_ is inverted due to the inverter. The inverted signal becomes non-inverted and occurs at TP_4_. Additionally, the clock generated at TP_1_ is inverted and generated at TP_3_. At this time, the signal speed control has the relationship (*R*_1_+*R*_2_)*C*_1_.

## 3. Manufacturing and Test Results

Figure 6 mimics the operating environment of the system with the designed TC and LUT algorithms. The *T_D_* data output from the TC is set as input to the LUT.

Figure 6 illustrates temperature change via a thermal imaging camera and temperature sensor using the designed module and phantom. The phantom was filled with the drug (normal saline: capacity −40 cc/beaker −100 mL) for the experiment. A fluid temperature 41–43 °C was introduced into the catheter (inflow part) of the phantom using the designed system. If the temperature in the catheter (outflow part) was 41–43 °C, the anticancer agent flowed into the catheter (inflow part). However, if the outflow temperature range was <41 °C, the system first heated the drug in the inflow until the drug achieved the desired temperature. Conversely, if the outflow temperature was >43 °C, the system stopped operating until the temperature fell within the desired temperature range.

To evaluate the operability of this algorithm, a temperature sensor and pulse waveform measurement sensor were installed to observe signal status through an external monitoring system. As illustrated in Figure 7a, an external monitoring system was used to measure the temperature and thermal compensation regulation. The external monitoring system can observe the pulse signal for the current drug temperature state and temperature decrease, increase, and maintenance states as illustrated in Figure 7b.

The pulse change process to generate *T_D_* when comparing *T_ref_* and *T_o_* is illustrated in Figure 7b. Here, the time required for pulse signal generation for one cycle change required for instantaneous heating, cooling, and maintenance is analyzed to be approximately 0.33 s. In other words, the size of the temperature changes over time, as shown in Figure 7a. For treatment, the temperature is maintained between 41 and 43 °C and the pulse changes, as shown in Figure 7b. If the temperature decreases to 38 °C, it increases to 43 °C through TC and LUT control and the pulse also changes. Additionally, if the temperature exceeds 44 °C, the TC and LUT are controlled to lower them to 41 °C and the pulse is changed accordingly.

When *T_o_* was 41–43 °C, *T_D_* output in the TC was 41–43 °C. This temperature corresponds to the *T_s_* signal (holding signal that can be introduced into the inflow). When *T_o_* was 38–40 °C, *T_D_* in the TC was < 41 °C. A waveform was obtained to increase the inflow temperature to 41–43 °C via *T_p_* to induce heating. When *T_o_* was 44–46 °C or higher, *T_D_* output in the TC was a temperature of 43 °C or higher. *T_a_* and *T_x_* were generated to stop system operation and enable cooling, and a waveform decreased the inflow temperature to 41–43 °C. To test the performance of the manufactured system, we used measuring equipment, as shown in Figure 7c. The measurement equipment environment consists of an oscilloscope (laptop (TBS1000C, TekScope TM, TEKTRONIX, INC., Seoul, Republic of Korea)), thermometer (GT521-Datalogger Ther-mometer, Gilwoo, Co., Ltd., Seoul, Republic of Korea), IR thermometer. (UT302C), and a thermal imaging camera (FLIR E5-Genius Industries, Seoul, Republic of Korea).

In particular, in the simulation result of Figure 7a, even if the temperature value changes in real time, the temperature level within the appropriate treatment range is adjusted in real time by the control command signals of the TC and LUT. As shown in Table 2, m_0_ is the initial temperature (room temperature) of the anticancer drug, and m_1_ is analyzed as the point where the anticancer drug begins to heat and the temperature rises. At this time, if the anticancer drug is heated and rises to m_2_, the TC and LUT are analyzed to increase the anticancer treatment effect by controlling the temperature from m_2_ to m_23_ to maintain the appropriate treatment temperature for 120 min.

Figure 8 shows that when *T_D_* generated by the TC is input to the LUT, the temperature range of 41–43 °C is maintained via heating and cooling by the order (ord) signal (*T_p_*, *T_a_*, *T_s_*, and *T_x_*) generated by the LUT to achieve the desired outcome. In addition, we have evaluated the performance of the system via a thermal imaging camera.

After designing the circuit using the ps-pice tool, as shown in Figure 5b, the results were obtained through simulation, as shown in Figure 7. At this time, it is assumed that the temperature (*T_o_*) of the drained anticancer agent in Figure 6b and the reference temperature (*T_ref_*) match (*T_D_* = 0 °C) through comparison and judgment (U2 in Figure 5b). If this happens, the temperature of the drug may not match during the process of being injected into the abdominal cavity and a difference may occur (*T_D_* = 1 °C). At this time, circuit operation can be induced by designing coding (Figure 6) so that a command signal can be sent from the LUT (U3 in Figure 5b) to heat or cool in the heater. This design process is presented in Figure 6, and the designed circuit was manufactured using a PCB board and measured using a phantom and a thermal imaging camera sensor as shown in Figure 8. The measurement results were compared to the *T_o_*/*T_D_* results in Table 1. Whenever the temperature changed (<41 °C, >43 °C), a phenomenon to maintain the treatment temperature (41–43 °C) could be observed. At this time, the problem of heterogeneity due to temperature change occurs. The cause of heterogeneity is a slight difference in the inflow temperature, outflow temperature, and intra-abdominal circulation temperature, resulting in a difference between the flow rate of the anticancer drug compared to the catheter space in the inflow and the flow rate of the anticancer drug compared to the catheter space in the outflow. It is believed that this is due to the difference in circulation flow rate compared to the area within the abdominal cavity. However, the distributed temperature is within the optimal treatment temperature range (41–43 °C), so it has been proven to be significant in clinical practice.

We conducted animal experiments to evaluate the effectiveness of the system. Animal testing is used to evaluate whether the system can be used clinically. To test the performance of the designed HIPEC surgery system, we conducted an animal trial. The animal was tested at the experimental animal center of HLB Biostep (Songdo Research Center, Incheon, Republic of Korea). We obtained institutional review board (IRB) permission from the animal ethics commission (BIOSTEP IACUC 23-KE-0515). In the animal test, a male farm mini pig (1 ea) weighing 35 kg was used. As shown in Figure 9a, the catheter and the HIPEC surgical system were connected through a tube, and filters were inserted into the ends of the inflow tube and the out-flow tube to block foreign substances flowing out of the abdomen.

The inflow catheter and the outflow catheter perform important functions in injecting and draining drugs into the abdomen and circulating the drugs. To inject drugs into the abdomen, we used normal saline instead of anticancer drugs to test the performance of the device. As shown in Figure 9a, we made an incision in the abdomen and opened it with a retractor. Measures were taken to enable inflow, outflow, and circulation by inserting a catheter into the stomach and sigmoid colon.

After catheterization, we sutured the abdomen so that only the tube was exposed on the outside of the abdomen and injected medication into the fluid box. After starting the system, the drug in the fluid box was heated to 44 °C, as shown in Figure 9b. The heating time was about 3 min.

The drug contained in the fluid box was administered into the abdomen through an inflow catheter. The drug circulates in the abdomen and is drained back to the outside through the outflow catheter. The drained drug was injected back into the abdomen through circulation. The temperature of the inflow catheter was 42.64 °C, as shown in Figure 9c, and the temperature of the output catheter was 41.71 °C.

We measured the temperature on the abdomen using a thermal imaging camera. The temperature of the measurement result was approximately 41.5–41.7 °C, as shown in Figure 9d. At this time, the temperature at which the fever dropped below 41 °C in the outflow catheter was 40.47 °C but reached 41.0 °C in 33 s through automatic control. Additionally, the temperature when the fever exceeded 43 °C was 44.49 °C. However, through automatic adjustment, the heat was reduced to 42.1 °C. At this time, it took 26 s to reach 42.1 °C. Therefore, the temperature could be maintained at 41–43 °C through automatic control, as shown in Figure 9d.

## 4. Discussion

If the temperature of anticancer drugs exceeds 46 °C, it induces damage to normal tissues [11,12,13,14,15,16,19]. If the temperature of anticancer drugs is below 41 °C, It cannot promote the inhibition of cancer tissue growth and rather reduces treatment performance [11,12,13,14,15,16,19]. However, if the anticancer drug is maintained at 41–43 °C, the growth promotion of cancer tissue is suppressed and treatment performance improves [11,12,13]. Therefore, the appropriate temperature for treatment is very important.

In order to distribute the anticancer drugs, which are at a temperature of 41 °C to 43 °C, evenly within the abdominal cavity, the staff repeatedly press and massage the abdomen with their hands. Additionally, as shown in Figure 10, the temperature distribution within the abdominal cavity can be checked using a thermal imaging camera, as shown in Figure 9, for each abdominal location (stomach, transverse colon, descending colon, ascending colon, sigmoid colon).

If the inflow temperature (anticancer drug) is 41 °C to 43 °C, when the anticancer drug flows into the abdominal cavity, the temperature may actually drop due to the flow rate and the space area within the abdominal cavity. Additionally, the temperature may not be 41 °C to 43 °C overall. In other words, the temperature may vary from location to location, so there may be locations where the temperature increases, locations where the treatment temperature is optimal, and locations where the temperature decreases treatment satisfaction. Therefore, the temperature should be distributed evenly through abdominal massage. At this time, the inflow must be maintained at least 43 °C. Therefore, the temperature distributed throughout the abdominal cavity through abdominal massage should be at least 41 °C, considering the phenomenon of temperature reduction.

The disadvantages and limitations identified in clinical practice are that the temperature of anticancer drugs may change frequently within the abdominal cavity, so it may be inconvenient to photograph the surface of the abdomen using a thermal imaging camera and observe the temperature value with the naked eye. At this time, in order to maintain the temperature at an appropriate temperature for treatment whenever the temperature changes, clinical sites must manually adjust the temperature level using human resources. Then, there is the continuous inconvenience of having to manually measure with the thermal imaging camera to determine the maintenance of the appropriate temperature for treatment within the abdominal cavity. Therefore, dedicated personnel are needed to observe the heat of the abdominal surface and to manually control the temperature, so the need for physical labor also increases. However, the proposed system detects the temperature in real-time, changes the real-time temperature through automatic control to maintain the changing temperature at an appropriate temperature level for treatment, and maintains convenient performance by displaying the current temperature value through the display. Therefore, it can reduce dedicated manpower and physical fatigue and is economically superior.

The focus of the LUT system design was to ensure that the data were accurate due to the coding work. So, the most important thing was that the signal provided by the TC was accurate. Delay (555 module in Figure 5b) occurs during the processing of signals provided by the TC (Q and TH signals in Figure 5b). Delay signal occurs for approximately 50 μs or more. However, the alternative we had overcome to minimize this delay phenomenon was to adjust the *R*_1_, *R*_2_, and *C*_1_ and *C*_2_ values (*C*_2_: tuning factor for fine adjustment), as shown in Figure 11a, using Equations (3)–(8). Through analysis, they were optimized to reduce delay. Therefore, the delay phenomenon was reduced to 0.22 μs by tuning the time constant.
(3)$$f=\frac{1.44}{\left(R_{1}+2R_{2}\right)C_{1}}$$
(4)$$T=\frac{1}{f}=0.639\left(R_{1}+2R_{2}\right)C_{1}$$
(5)$$t_{on}=0.693\left(R_{1}+R_{2}\right)C_{1}$$
(6)$$t_{off}=0.693R_{2}C_{1}$$
(7)$$T_{D}=t_{on}+t_{off}$$
(8)$$D=\frac{t_{on}}{T_{D}}=\frac{R_{1}+R_{2}}{R_{1}+2R_{2}}\times 100\%$$

*The R*_1_ and *R*_2_ and *C*_1_ and *C*_2_ values are provided in Figure 5b. In the equation, *f*, *T*, *t_on_*, *t_off_*, *T_D_*, and *D* are the frequency, cycle, circulation time of the anticancer drug, temperature conversion time, and temperature conversion time difference compared to the anticancer drug purification time (compared to the time required to restart after the temperature is converted), respectively. It has the meaning of a time scale to reduce delay phenomenon and the ratio of the time it takes for the circulation time and temperature to be converted (duty cycle). Therefore, *f*, *T*, *t_on_*, *t_off_*, *T_D_*, and *D* are 0.017 Hz, 58.6 s, 45.6 s, 13.03 s, 58.68 s, and 77.8%, respectively. Here, if the temperature deviates from the appropriate treatment temperature range during the process of maintaining the temperature, heat has the meaning of rapidly changing the temperature through instantaneous cooling and heating. Therefore, as shown in Figure 11b, the shorter the *T_off_*, the shorter the time required to change the temperature, and the larger the duty cycle, the better the temperature conversion performance. This tuning process took a considerable amount of time, and the tuning results require that the signal provided by the TC be fast and accurate, so it is analyzed that continued research will be needed to further shorten toff in the future. In other words, research to increase the time constant can contribute to improving the surgical performance by improving the reliability and speed of system operation and shortening the surgical time because the signal provided by the LUT is accurate.

The aim of this study, which was a non-clinical trial, was to evaluate the accuracy and effectiveness of TC of the proposed module. Therefore, heat/temperature controllability was evaluated. Currently, drug temperature during HIPEC treatment is manually controlled by an assistant. If the temperature of the drug rises, the input heat is lowered, and vice versa. In addition, if fever is constant, the system is on standby. However, if fever decreases or increases, the treatment is stopped. Consequently, the treatment is cumbersome and time consuming. However, this study has the advantage of automatically adjusting the heat through temperature monitoring to solve the cumbersome process.

If a *T_s_* signal is generated in the LUT, the anticancer agent flows into the inflow. However, if a *T_p_* signal is generated, the heat generator increases output. Conversely, if *T_a_* or *T_x_* signals are output, the heat generator will stop operation. In both cases, the anti-cancer drug flows through the inflow only when a *T_s_* signal is generated.

When a drug at a temperature of 43 °C is injected into the abdominal cavity through the inflow, the local tissue temperature in the abdominal cavity decreases due to the first law of thermodynamics (thermal equilibrium). If the temperature of the drug flowing out of the abdominal cavity is 41 °C, the local tissue temperature in the abdominal cavity is estimated to be less than 41 °C. Therefore, considering that the temperature of the drug will decrease, the heater is often set to 46 °C (temperature that normal tissues can tolerate) when injecting anti-cancer drugs into the peritoneal cavity [26]. Consequently, the set temperature of 46 °C will decrease to 43 °C during intraperitoneal injection. When a drug at a temperature of 43 °C is injected intraperitoneally, the drug temperature at the outflow is less than 41 °C due to heat reduction, and the reduced temperature (*T_o_*) is rapidly compared with *T_ref_* by the TC to obtain the TD. This TD value may be higher or lower than 41 °C.

Heat control methods (instantaneous heating) using temperature monitoring and control in HIPEC surgery include hot water and microwave (or high-frequency induction local heating thermal treatment) heating [26,27,28,29,30,31,32,33,34]. These methods suffer from low precision and unstable perfusion rate during temperature control [27]. Because maintaining stable heat input for at least 60 min during system operation is difficult, a method of controlling the temperature with continuous monitoring and proper control is essential [35].

When the drug flows into the catheter after the desired temperature is achieved, the temperature in the abdominal cavity is monitored at 40.5–41.5 °C with a set temperature of 45.3 °C. Since the intraperitoneal temperature must be maintained between 41 and 43 °C, precise control technology within ±5% is required by rapidly detecting a decrease or increase in heat. Accordingly, steady temperature control through instantaneous resistance heating is a desirable solution that provides a control accuracy within ±0.5% [36].

The latest developments include a plastic tube heating element, a temperature control device, and a temperature sensor for instantaneous heating and temperature control [37]. To improve the precision of temperature control, efficiently controlling heat by comparing and judging a fitting result of mathematical modeling and a monitoring signal was developed. The control technology used a proportional–integral–differential controller (PID) with at least 95% accuracy and ±5% tolerance range [27]. Here, the time required by the TC module to determine *T_D_* and *T_ref_* is critical [35]. If *T_D_* < 41 °C (for example, 38 °C), the TC must generate a fast control signal, as shown in Figure 12, to maintain desired inflow temperature For example, when *T_o_* is 38 °C, the time (*T_PR_*) required to regulate *T_D_* to 41 °C is 0.89 s. When *T_o_* is 41–43 °C, *T_D_* is 41–43 °C as well. The time (*T_R_*) required by the TC to determine this condition is 0.40 s. Assuming that the anti-cancer agent is maintained at 0 °C before surgery, the time (*T_P_*) required for instantaneous heating is approximately 0.11 s. As shown in Equations (9) and (10), evaluating these control times to obtain the desired *T_D_* value is necessary [35]. During evaluation, values of *T_P_*, *T_PR_*, and *T_R_* closer to 0 s indicate superior performance. Further, *T_P_*, *T_PR_*, and *T_R_* should be designed such that they do not exceed 1.0 s. Therefore, *T_R_*, *T_PR_*, and *T_P_* should be close to 1/10 level. The values for *T_P_*, *T_PR_*, and *T_R_* must theoretically be linear. However, they have a non-linear slope. Therefore, *T_P_*, *T_PR_*, and *T_R_* can be determined mathematically using the overlapping points of the linear and nonlinear slopes [35]. Analysis of *T_P_*, *T_R_*, and *T_PR_* for excellent control performance (closer to 0 s) can be judged through Equations (9) and (10). A value closer to 0 s is ideal. Accordingly, the average time (*A_PR_*) required for control from *T_P_* to *T_PR_* is 0.11 s and control (*A_RR_*) is 0.45 s. *A_PR_* and *A_RR_* in [10] are 0.225 and 0.6781 s, respectively, and the *A_PR_* and *A_RR_* of the proposed system are 0.11 and 0.45 s, respectively. Therefore, the *A_PR_* was more than 2 times better than [27], and *A_RR_* was more than 1.51 times better.
(9)$$A_{PR}=\frac{T_{P}}{T_{PR}}$$
(10)$$A_{RR}=\frac{T_{R}}{T_{PR}}$$

Through Equation (11), we analyzed the steady-state temperature (*t_ss_* = 43.0), offset temperature (*t_o_* = 38.0), steady state of the input signal (*u_ss_* = 40.0), and input signal offset (*u_o_*) to determine heating system operation. *K_HS_* was analyzed for assessing the sensing capabilities of the TCs, where *u_o_* is 0 (*u_o_* = 0). A *K_HS_* value closer to 0 indicates better performance. Accordingly, the *K_HS_* obtained from Equation (11) was 0.125.
(11)$$K_{HS}=\frac{t_{ss}-t_{o}}{u_{ss}-t_{o}}$$

Even if the heat distribution changes through real-time temperature measurement, this system can quickly change the temperature to within an appropriate level for treatment through control of TC and LUT, as shown in Table 3. Therefore, assuming that T_1_ is the temperature at which the temperature begins to increase when the anticancer drug is heated, T_2_ is the temperature level in the process where the anticancer drug changes to a high temperature. At this time, the range from T_3_ to T_6_ is the temperature distribution area at an appropriate level for treatment. T_6_ to T_17_ can be viewed as control areas to maintain the appropriate temperature for treatment.

Table 4 records the comparison and evaluation results for the difference between the *K_HS_* of the proposed and existing methods. During evaluation [37,38,39,40], the *K_HS_* was observed to improve by more than two times.

As shown in Figure 5, Figure 6, and Figure 8, the performance effectiveness of the system was evaluated through design, simulation results, manufacturing, and measurement. To verify the validity of these performance evaluation results, we performed animal experiments. In animal experiments, changes in temperature were observed using a thermal imaging camera. Measurements were attempted to evaluate temperature changes and control performance effectiveness at the surgical site, and the results (Figure 9) were obtained.

In the future, we plan to conduct clinical trials through medical device licensing and pursue the advancement of the system by investigating matters that need to be supplemented through clinical trials. To this end, we plan to analyze the inflow and outflow of anticancer drugs, the flow rate for circulation through finite element analysis, the flow rate for volume (inflow/outflow catheter and volume within the abdominal cavity) through spatial coordinate analysis, and the flow rate that occurs according to the flow rate. Analysis of heat distribution and changes will be required. In order to ensure smooth inflow and outflow of anticancer drugs into the abdominal cavity and circulation through the catheter, motor performance (suitability of power and speed), thermal noise generated from the motor, motor heating and cooling performance, overload reduction technology, and electrical devices that interfere with signal detection interference factors and electrical interference suppression technologies will need to be comprehensively analyzed. Expert advice is essential for an electromagnetic compatibility assessment, and such work is essential to consider in medical device licensing. In order to carry out these tasks smoothly, it is necessary to conduct additional animal experiments and then pursue a licensing plan through consultation with medical device licensing experts, obtain approval, and conduct clinical trials. Additional animal experiments include growing cancer cells in nude mice, transplanting them into the pig colon, and connecting the designed system to the pig’s abdomen through a catheter to inject, drain, and circulate high-temperature anticancer drugs to damage cancer cells (possibility of inhibiting HSP90 growth) and surrounding areas. We plan to evaluate tissue survival and the effectiveness of real-time temperature detection and control to maintain an appropriate temperature for treatment. We plan to contribute to obtaining results by recruiting patients from control and experimental groups during the clinical trial. During the clinical trial process, we plan to compare the existing system and the proposed design system to test the speed of temperature judgment and comparison, temperature control ability, and real-time temperature detection response speed. Additionally, we plan to apply graph technology to increase the instantaneous heating speed. After completing clinical trials, it is expected that technology transfer or commercialization will be possible if positive results are obtained after registering a level 3 medical device license with the Ministry of Food and Drug Safety and demonstrating use at a medical institution. After conducting trial use at major medical institutions to investigate matters that need to be improved, efforts should be made to continuously improve performance to enable sufficient use in clinical settings. In particular, the unit price and maintenance costs of the HIPEC surgical system are high because it relies on imported products. Even if the performance is disappointing, since it is an imported product, we have no choice but to adapt and use it. However, if continuous development is achieved through domestically produced technology, entry barriers can be overcome in terms of unit cost, maintenance, and economics. In particular, if maintenance (A/S) services are improved, it is expected that distribution will be smooth and that it will play a leading role in leading medically advanced countries at the level of international competition. In addition, if the product is delivered through a survey of customer requirements and desired consumer preferences, the proposed system technology is expected to increase continuous development because it can be supplied to medical institutions around the world through domestic and overseas sales.

Since the proposed system is an HIPEC surgical system, it can be used to treat colon cancer, ovarian cancer, and peritoneal cancer, so its utilization is expected to increase in the future through research and development and demand for use in medical settings is also expected to increase [41,42,43].

## 5. Conclusions

Currently, drug temperature during HIPEC is manually controlled by an assistant. Therefore, the treatment process is cumbersome and time-consuming. The proposed TC method in this study automatically adjusts heat input via temperature monitoring to facilitate temperature control.

The method of comparing drug temperature, collecting and analyzing LUT data, and maintaining and compensating for temperature through control signal output is a very important and groundbreaking technology.

Existing HIPEC devices are equipped with a monitoring device for changes in temperature. However, there is no automatic control device that can automatically change the temperature to the appropriate treatment temperature whenever the temperature changes. The novelty of this research and development is that it has the ability to detect temperature in real time and control the temperature. This function has the advantage of allowing safe treatment and improving treatment results by automatically adjusting the temperature to the appropriate treatment temperature range whenever the temperature changes in real time. This experiment was conducted on animals by connecting the fluid box, the designed circuit, and the monitor as a single unit. Future plans will be to integrate the proposed design circuit, a fluid box for momentarily heating anticancer drugs, a perfusion filter, and a monitor and to conduct animal and clinical tests to obtain excellent results in operation. We plan to study a design method for instantaneously heating anticancer drugs in a fluid box using graph materials.

Thus, animal experiments will be necessary for the commercialization of this system in the future. The proposed method can reduce errors due to manual temperature control methods, guarantee accurate and safe treatment, and prompt surgery via automatic control functions. The proposed system can be applied to surgical treatments in surgery, obstetrics, and gynecology.

## Figures and Tables

**Figure 1 sensors-24-00596-f001:**
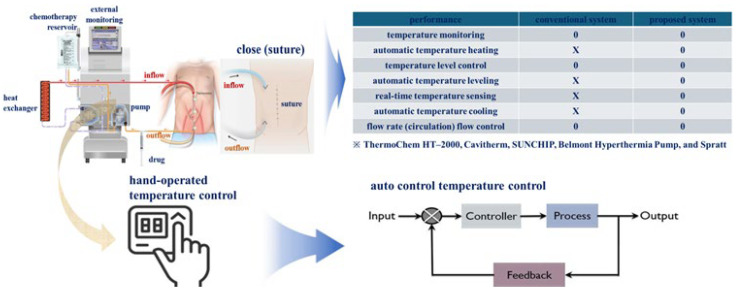
Comparison of characteristics of existing and proposed systems and overview of research purpose.

**Figure 2 sensors-24-00596-f002:**
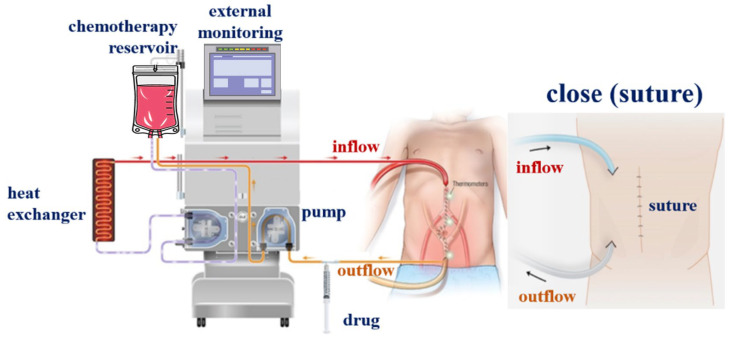
HIPEC surgery system schematic.

**Figure 3 sensors-24-00596-f003:**
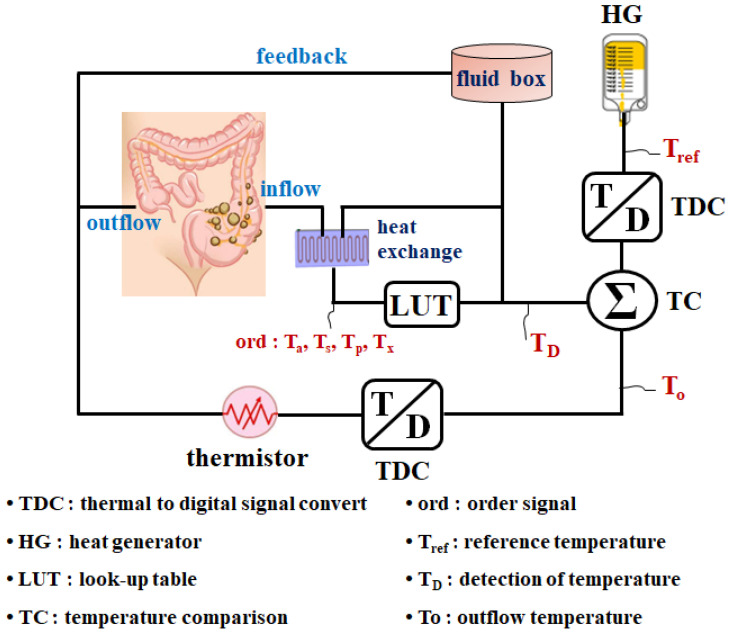
Proposed thermal control system with TC and LUT.

**Figure 4 sensors-24-00596-f004:**
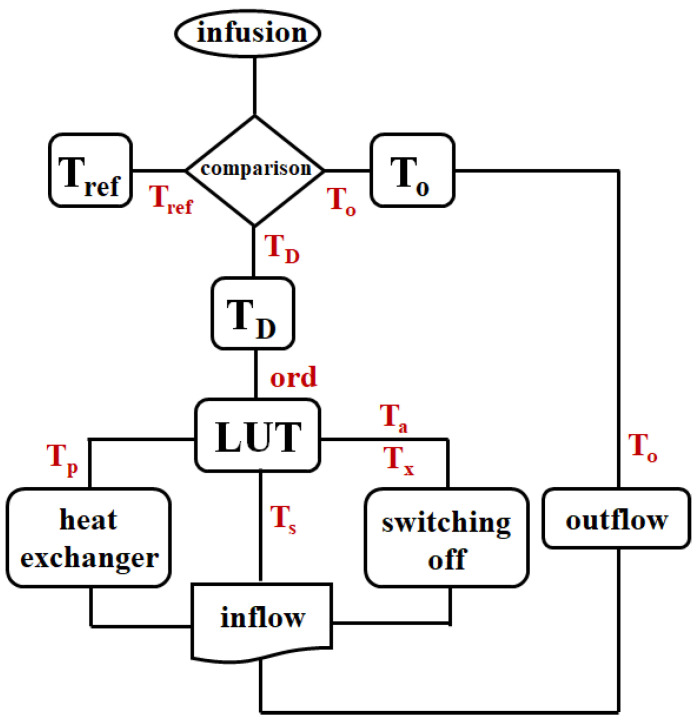
Flow chart of the proposed system with temperature control.

**Figure 5 sensors-24-00596-f005:**
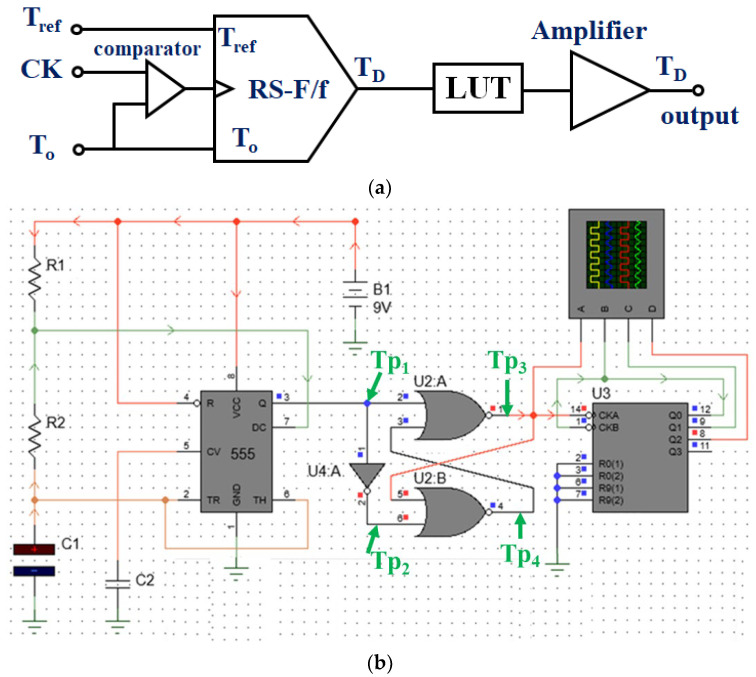
Proposed structure of the HIPEC system with temperature control: (**a**) block diagram and (**b**) schematic (*R*_1_: 10 MΩ, *R*_2_: 4.0 MΩ, *C*_1_:4.7 μF, *C*_2_: 0.1 μF) (**c**) simulation results for schematic with temperature control (see Appendix A for algorithm coding data in the web site).

**Figure 6 sensors-24-00596-f006:**
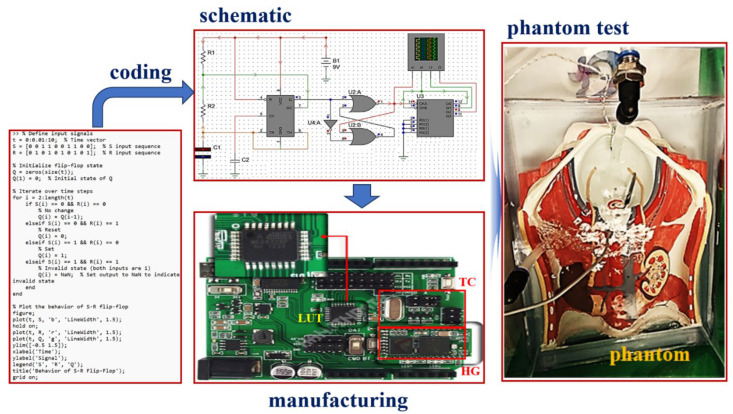
Implementation of the algorithm (coding) and phantom evaluation (see Appendix A for algorithm coding data in the web site).

**Figure 7 sensors-24-00596-f007:**
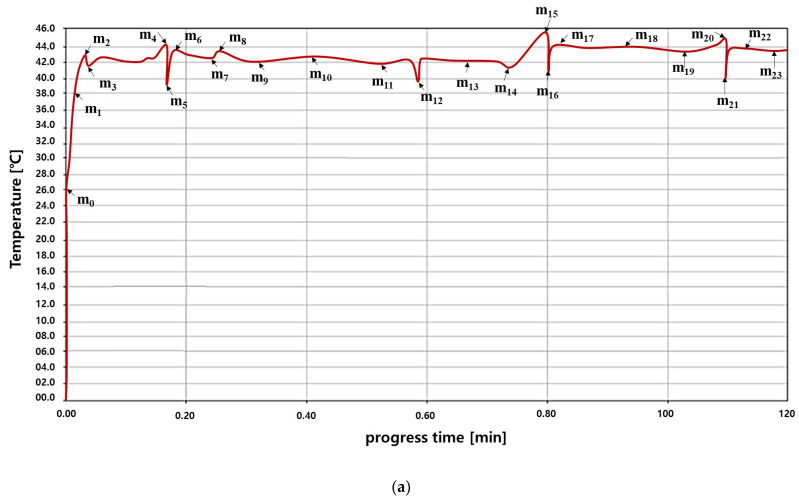
Measurement results for proposed TC processing: (**a**) TC temperature, (**b**) pulse signal with TC, (**c**) system performance measurement system environment space.

**Figure 8 sensors-24-00596-f008:**
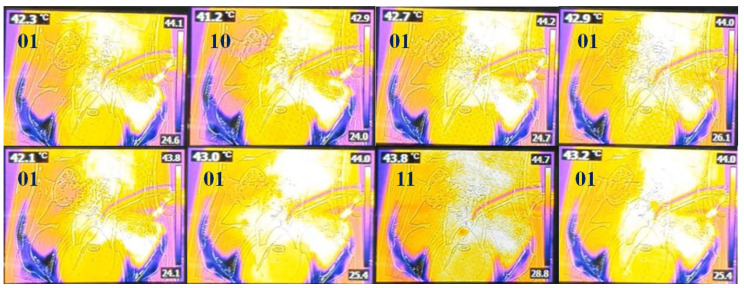
Thermal TC images using the LUT to assess heating and cooling performance (01: *T_a_*, 10: *T_p_*, 11: *T_x_* according to the Table 1 of LUT = *T_ref_* // *T_o_*).

**Figure 9 sensors-24-00596-f009:**
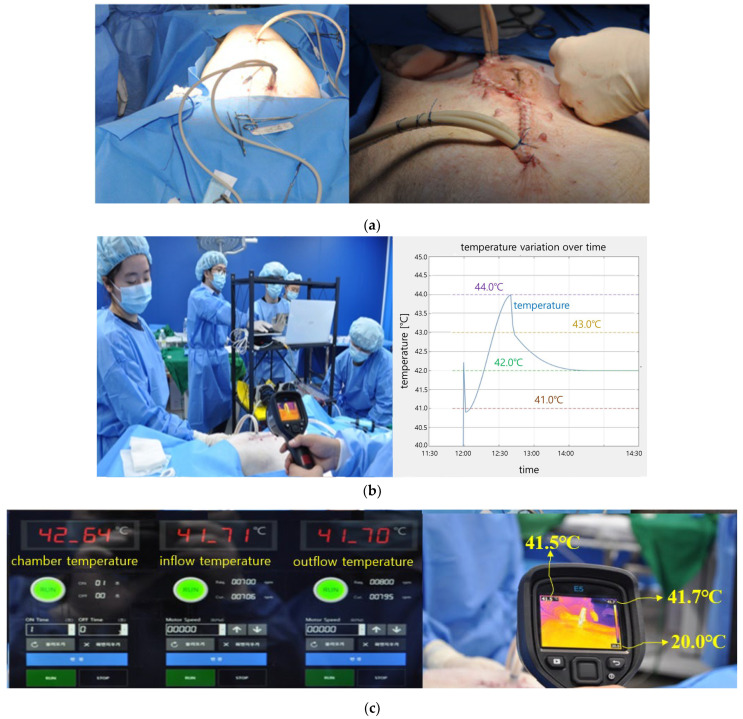
Temperature control and thermal measurement results of the HIPEC surgical system through animal experiments. (**a**) Surgery and catheter tube docking, (**b**) temperature of the heat generation, (**c**) measurement result of the temperature for heat generation and inflow part, (**d**) measurement results for abdomen temperature.

**Figure 10 sensors-24-00596-f010:**
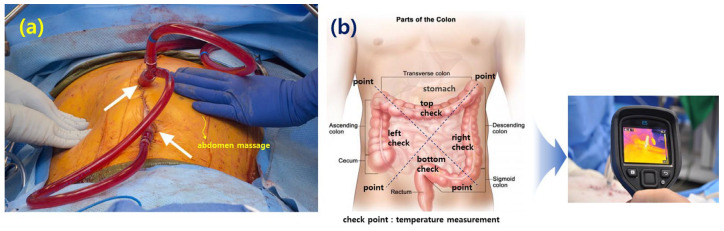
Temperature distribution and distribution temperature check method in HIPEC surgery. (**a**) Abdominal massage [15], (**b**) colon point temperature check.

**Figure 11 sensors-24-00596-f011:**
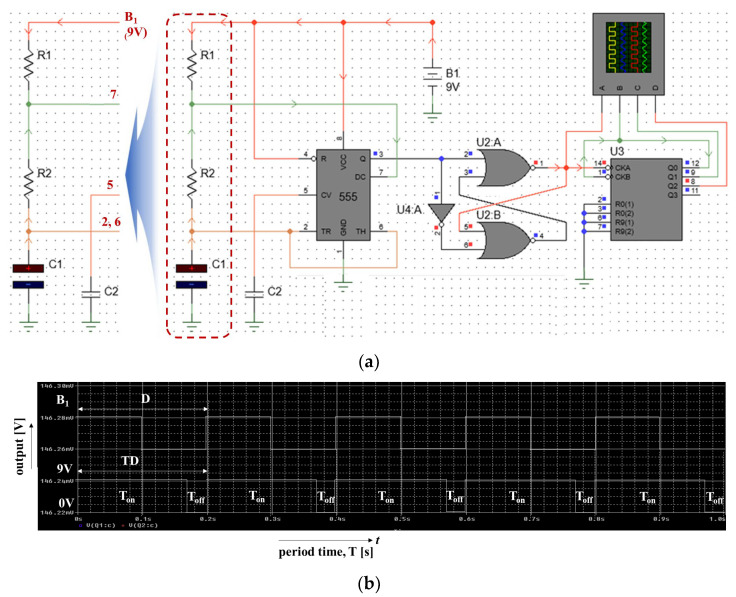
Time constant adjustment to reduce temperature control time. (**a**) Time constant adjustment, (**b**) time constant adjustment response characteristics.

**Figure 12 sensors-24-00596-f012:**
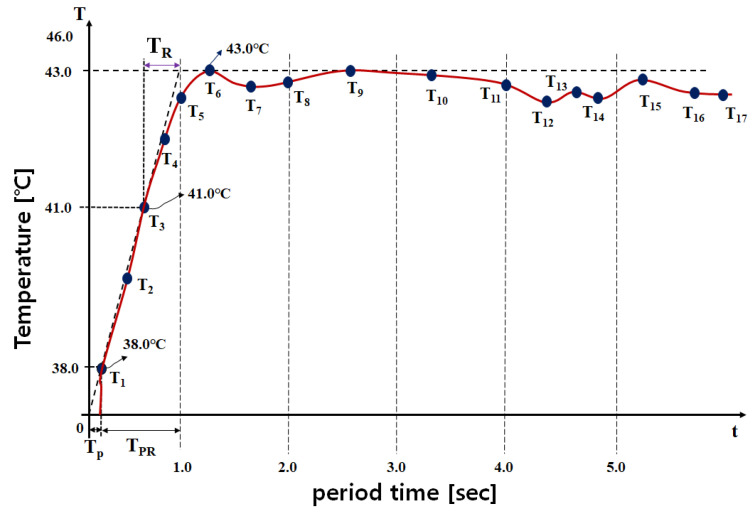
Performance test graph for the TC unit (T: temperature and t: period time).

**Table 1 sensors-24-00596-t001:** Suggestion of the LUT.

Temperature Range	Signal to LUT			Status of Temperature	
	*T_ref_*	*T_o_*	*T_D_*	ord	LUT
<40 °C	1	0	1	up	*T_p_*
41–43 °C	0	0	1	stead	*T_s_*
>44 °C	0	1	0	down	*T_a_*
>45 °C	1	1	1	prohibited to use	*T_x_*

**Table 2 sensors-24-00596-t002:** Variation of the temperature distribution area by TC and LUT control.

Performance	Temperature [°C]	Performance	Temperature [°C]
m_0_	26.0	m_12_	39.9
m_1_	38.0	m_13_	42.0
m_2_	43.0	m_14_	41.7
m_3_	41.8	m_15_	45.7
m_4_	44.0	m_16_	40.8
m_5_	38.3	m_17_	40.7
m_6_	42.9	m_18_	43.9
m_7_	42.2	m_19_	43.5
m_8_	43.0	m_20_	44.4
m_9_	42.0	m_21_	40.0
m_10_	42.2	m_22_	43.9
m_11_	41.9	m_23_	43.4

**Table 3 sensors-24-00596-t003:** Temperature distribution area by TC and LUT control.

Performance	Temperature [°C]	Characteristic
T_1_	38.0	anticancer drug room temperature
T_2_	40.5	Initial heating temperature of heat generator
T_3_	41.0	Minimum temperature range suitable for treatment
T_4_	41.5	Treatment temperature range
T_5_	42.5	Treatment temperature range
T_6_	43.0	Treatment temperature range
T_7_	42.7	Treatment temperature range
T_8_	42.8	Treatment temperature range
T_9_	43.0	Treatment temperature range
T_10_	42.9	Treatment temperature range
T_11_	42.8	Treatment temperature range
T_12_	42.4	Treatment temperature range
T_13_	42.6	Treatment temperature range
T_14_	42.4	Treatment temperature range
T_15_	42.8	Treatment temperature range
T_16_	42.5	Treatment temperature range
T_17_	42.4	Treatment temperature range

**Table 4 sensors-24-00596-t004:** Comparison of the performance for proposed and existing systems.

Ref [#]	*t_ss_* [°C]	*t_o_* [°C]	*u_ss_* °C	*K_HS_*	Times [xN]	Temperature Control Method
this work	43.0	38.0	40.0	0.125	–	RS–F/f LUT
[37]	19.2	15.0	15.5	0.271	2.17	PID control
[38]	19.2	15.0	15.5	0.271	2.17	IMC discrete feedback controller
[39]	20.0	4.00	43.7	1.968	15.7	Feed-forward control
[40]	10.0	0.00	23.5	0.425	3.40	Internal Model Control (IMC)

## Data Availability

The data presented in this study are available upon request from the corresponding author. The data are not publicly available because of privacy and ethical restrictions.

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
