# Peer review of "Comparative Sensing and Judgment Control System for Temperature Maintenance for Optimal Treatment in Hyperthermic Intraperitoneal Chemotherapy Surgery"

_sensors, 2024, doi:10.3390/s24020596_

Round 1
Reviewer 1 Report
Comments and Suggestions for Authors
This article proposes a comparator and look–up table (LUT) to maintain drug temperature within the desired range during intraperitoneal circulation during hyperthermic intraperitoneal chemotherapy. The proposed method automatically adjusts heat input via temperature monitoring to facilitate temperature control. The manuscript is interesting, proposed approach has undoubted practical significance.
However, article is currently not suitable for receiving, and some modifications need to be made before it can be accepted for publication.
- In Introduction authors mentioned “Normal tissue can withstand temperatures up to 46 °C, while tumors necrotize be-41 tween 41–43 °C.” References confirming this statement must be provided.
- Figure 7. Thermal TC images using the LUT to assess heating and cooling performance.Please explain what is shown in the picture? What causes the heterogeneity of temperature distribution? Was the temperature indicated in the legend measured at the indicated point? It is also necessary to provide a color scale demonstrating the correspondence between temperature and color in the figure.
-Lines 190-199. Please check the line spacing.
-Please check the℃ symbol font, it seems different from the rest of the text.
- Figure 6 and Figure 9 must be improved. Please change the color of the axes and text to black. It is also necessary to show the experimental points and STD in both plots.
Author Response
- Question
This article proposes a comparator and look–up table (LUT) to maintain drug temperature within the desired range during intraperitoneal circulation during hyperthermic intraperitoneal chemotherapy. The proposed method automatically adjusts heat input via temperature monitoring to facilitate temperature control. The manuscript is interesting, proposed approach has undoubted practical significance.
However, article is currently not suitable for receiving, and some modifications need to be made before it can be accepted for publication.
- answers
Thank you for your thorough review of the paper. We have done our best to reflect your opinion.
- Question
In Introduction authors mentioned “Normal tissue can withstand temperatures up to 46 °C, while tumors necrotize be-41 tween 41–43 °C.” References confirming this statement must be provided.
- answers
HIPEC surgery maintains the temperature at 41℃ to 43℃ by injecting high-temperature anticancer drugs into the abdominal cavity for 90-120 minutes. The reason is that normal tissue is damaged at temperatures above 46℃, while cancerous tissue becomes necrotic at 41℃ to 43℃. Please refer to lines of 70-777 (green) for these sentences. The evidence is clear in references [11]-[16], [19]. Additionally, the reason cancer cells are damaged by heat between 41℃ and 43℃ is because HSP90 growth is inhibited [17, 18]. Please refer to Lines of 65-69 (red).
Also, please refer to the lines of 323-328 (yellow) in the discussion.
If the temperature of anticancer drugs exceeds 46℃, it induces damage to normal tissues [11]-[16], [19]. If the anticancer agent is used at temperatures below 41°C, the inhibition and promotion of cancer tissue growth cannot be achieved. Rather, treatment performance deteriorates [11]-[16], [19]. However, if the anticancer drug is maintained at 41℃-43℃, the growth promotion of cancer tissue is suppressed and treatment performance improves [11]-[13]. Therefore, the appropriate temperature for treatment is very important. Therefore, the appropriate temperature for treatment is very important.
- Question
Figure 7. Thermal TC images using the LUT to assess heating and cooling performance. Please explain what is shown in the picture? What causes the heterogeneity of temperature distribution? Was the temperature indicated in the legend measured at the indicated point? It is also necessary to provide a color scale demonstrating the correspondence between temperature and color in the figure.
- answers
Thank you I have changed the test results and added a description to Figure 8 and the lines of 256-275 (red).
- Question
Lines 190-199. Please check the line spacing.
- answers
I adjusted the line spacing throughout. Thanks for pointing this out.
- Question
Please check the℃ symbol font, it seems different from the rest of the text.
- answers
All fonts have been matched. Thank you.
- Question
Figure 6 and Figure 9 must be improved. Please change the color of the axes and text to black. It is also necessary to show the experimental points and STD in both plots.
- answers
Changed it to black. Also, please make sure that your comments are written as intended.
Here's what you need to check:
Figure 7(a), Table 2, lines of 238-245 (yellow), Figure 12, Table 3, lines of 473-480 (yellow).
Reviewer 2 Report
Comments and Suggestions for Authors
Reviewer's comments:
Before publication, the authors have to correct the typos that are found all over the manuscript and address the followings:
1. The Introduction should consist of five paragraphs answering the following five questions: What is the problem? Why is it interesting and important? Why is it hard? Why hasn't it been solved before? (or, what's wrong with previously proposed solutions? What are the key components of my approach and results?
2. What is the novelty in your work, please explain?
3. What is the rationale behind the specific temperature range (41–43 °C) chosen for thermal therapy in the proposed system?
4. How critical is it to maintain the drug temperature within this range for effective treatment?
5. What are the limitations and challenges associated with the existing HIPEC devices mentioned (ThermoChem HT–2000, Cavitherm, SUNCHIP, Belmont Hyperthermia Pump)?
6. How do these commercial devices handle temperature distribution in the abdominal cavity, and what are their drawbacks?
7. What were the shortcomings of Spratt's system, and how does the proposed system address those issues?
8. Are there any anticipated challenges or limitations in implementing the proposed comparator and LUT system in a clinical setting?
9. Has the proposed system been experimentally tested or validated? If so, can you provide insights into the results and the system's performance?
10. Are there plans for further experimentation or clinical trials to validate the effectiveness of the proposed system?
11. What are the future directions for research or improvements in the proposed system?
12. Could you provide more details about the animal experiments mentioned for the future commercialization of the system? What parameters will be evaluated in these experiments?
13. Are there any specific technological aspects or components that differentiate this proposed system from existing temperature control methods in surgical treatments?
14. How scalable is the proposed system, and what considerations have been made to ensure its adaptability to various clinical settings?
15. What steps or plans are envisioned for the commercialization of this system, considering the need for animal experiments and subsequent regulatory approvals?
16. Are there any potential barriers or challenges anticipated in the commercialization process, and how do you plan to address them?
17. Can you provide insights into potential future research directions or improvements that could be made to enhance the proposed TC method?
18. Conclusions should be more concrete, and future research directions presented.
19. Overall, the work is interesting; it just needs to follow the suggestions to improve the manuscript.
Comments on the Quality of English LanguageReviewer's comments:
Before publication, the authors have to correct the typos that are found all over the manuscript and address the followings:
1. The Introduction should consist of five paragraphs answering the following five questions: What is the problem? Why is it interesting and important? Why is it hard? Why hasn't it been solved before? (or, what's wrong with previously proposed solutions? What are the key components of my approach and results?
2. What is the novelty in your work, please explain?
3. What is the rationale behind the specific temperature range (41–43 °C) chosen for thermal therapy in the proposed system?
4. How critical is it to maintain the drug temperature within this range for effective treatment?
5. What are the limitations and challenges associated with the existing HIPEC devices mentioned (ThermoChem HT–2000, Cavitherm, SUNCHIP, Belmont Hyperthermia Pump)?
6. How do these commercial devices handle temperature distribution in the abdominal cavity, and what are their drawbacks?
7. What were the shortcomings of Spratt's system, and how does the proposed system address those issues?
8. Are there any anticipated challenges or limitations in implementing the proposed comparator and LUT system in a clinical setting?
9. Has the proposed system been experimentally tested or validated? If so, can you provide insights into the results and the system's performance?
10. Are there plans for further experimentation or clinical trials to validate the effectiveness of the proposed system?
11. What are the future directions for research or improvements in the proposed system?
12. Could you provide more details about the animal experiments mentioned for the future commercialization of the system? What parameters will be evaluated in these experiments?
13. Are there any specific technological aspects or components that differentiate this proposed system from existing temperature control methods in surgical treatments?
14. How scalable is the proposed system, and what considerations have been made to ensure its adaptability to various clinical settings?
15. What steps or plans are envisioned for the commercialization of this system, considering the need for animal experiments and subsequent regulatory approvals?
16. Are there any potential barriers or challenges anticipated in the commercialization process, and how do you plan to address them?
17. Can you provide insights into potential future research directions or improvements that could be made to enhance the proposed TC method?
18. Conclusions should be more concrete, and future research directions presented.
19. Overall, the work is interesting; it just needs to follow the suggestions to improve the manuscript.
Author Response
- Question
Before publication, the authors have to correct the typos that are found all over the manuscript and address the followings:
- answers
Thank you for your thorough review of the paper. We have done our best to reflect your opinion.
- Question
The Introduction should consist of five paragraphs answering the following five questions: What is the problem? Why is it interesting and important? Why is it hard? Why hasn't it been solved before? (or, what's wrong with previously proposed solutions? What are the key components of my approach and results?
- answers
After reading the introduction, I understood the reviewer's question, so I revised and supplemented the sentences in the introduction. I've modified and supplemented it based on the questions you asked. Thanks for the good point. Please check the entire introduction sentence.
For the last question (key components), please refer to the lines of 70-92 (green and sky-blue) in the introduction.
- Question
What is the novelty in your work, please explain?
- answers
Existing HIPEC devices are equipped with a monitoring device for changes in temperature. However, there is no automatic control device that can automatically change the temperature to the appropriate treatment temperature whenever the temperature changes. The novelty of this research and development is that it has the ability to detect temperature in real time and control the temperature. This function has the advantage of allowing safe treatment and improving treatment results by automatically adjusting the temperature to the appropriate treatment temperature range whenever the temperature changes in real time.
This novelty is recorded in lines of 101-104 (green) and Figure 1 and in the conclusion of session 5, lines of 556-563 (green).
- Question
What is the rationale behind the specific temperature range (41–43 °C) chosen for thermal therapy in the proposed system?
- answers
HIPEC surgery maintains the temperature at 41℃ to 43℃ by injecting high-temperature anticancer drugs into the abdominal cavity for 90-120 minutes. This is because normal tissue is damaged above 46℃, but cancer tissue dies between 41℃ and 43℃ because HSP90 growth is inhibited. These sentences are reflected in lines of 65-69 (red) and lines of 70-77 (green) in the introduction, and the basis is clear in references [17, 18], [11]-[16], and [19].
- Question
How critical is it to maintain the drug temperature within this range for effective treatment?
- answers
This is because cancer tissue inhibits HSP90 growth between 41℃ and 43℃.
Detailed evidence is reported in references [17, 18] and changes in cell survival curves with temperature are documented in detail [17].
If the temperature of anticancer drugs exceeds 46℃, normal tissues are damaged [11]-[16], [19]. If the temperature of anticancer drugs is below 41℃, cancer tissue is not damaged and treatment performance deteriorates [11]-[16], [19]. However, if anticancer drugs are maintained at 41℃-43℃, cancer tissue promotes growth inhibition [11]-[13]. Therefore, the appropriate temperature for treatment is very important. Please refer to the lines of 323-328 (yellow) in the discussion.
- Question
What are the limitations and challenges associated with the existing HIPEC devices mentioned (ThermoChem HT–2000, Cavitherm, SUNCHIP, Belmont Hyperthermia Pump)?
- answers
Please refer to the explanation of lines of 77-92 (sky blue) in the introduction.
- Question
How do these commercial devices handle temperature distribution in the abdominal cavity, and what are their drawbacks?
- answers
You can check the lines of 323-363 (red) and Figure 10 in the discussion of Session 4. In order to distribute the anticancer drug, which is at a temperature of 41℃ to 43℃, evenly within the abdominal cavity, the abdomen is pressed and massaged with the operator's hands. In addition, as shown in Figure 10, a thermal imaging camera is used for each abdominal location (stomach, transverse colon, descending colon, ascending colon, sigmoid colon) as shown in Figure 9 (c) and Figure 9 (d). Therefore, the temperature distribution within the abdominal cavity can be checked.
In addition, referring to the lines of 77-92 (sky-blune) in the introduction, the operating status of the system can be monitored and the temperature must be physically adjusted directly to maintain the appropriate temperature for treatment while observing the temperature through a sensor. However, it is not easy to automatically adjust the optimal treatment temperature for intra-abdominal temperature distribution [20]. In particular, automatic sensors that can detect real-time temperature, devices for comparison and judgment to maintain the appropriate temperature range for treatment (41℃ to 43℃), and automatic temperature control devices are lacking.
- Question
What were the shortcomings of Spratt's system, and how does the proposed system address those issues?
- answers
In the introduction, you can see the lines of 93-97 (yellow), lines of 97-101 (black), and lines of 101-104 (green). In particular, please refer to the lines of 97-101 (black).
- Question
Are there any anticipated challenges or limitations in implementing the proposed comparator and LUT system in a clinical setting?
- answers
Please refer to notes 364-372 (pink) and Figure 11 in the discussion.
- Question
Has the proposed system been experimentally tested or validated? If so, can you provide insights into the results and the system's performance?
- answers
As shown in Figures 5, Figure 6, and Figure 8, the performance effectiveness of the system was evaluated through design, simulation results, manufacturing, and measurement. In particular, the experimental test results for Figure 5 can be obtained through Figure 5(c). Figure 5(c) can be obtained through simulation.
For details, please refer to the lines of 170-175 (red).
To concretely verify the verification, measurements were made using measuring equipment as shown in Figure 7 (c). Please check the lines of 223-228 (sky blue) for the measurement equipment environment configuration.
In addition, as shown in the lines of 256-275 (red), the circuit was designed using the ps-pice tool as shown in Figure 5 (b), and the results were obtained through simulation as shown in Figure 6. At this time, in Figure 7 (b), if the temperature (To) of the drained anticancer drug and the reference temperature (Tref) match (TD = 0°C) through comparison and judgment (U2 in Figure 5 (b)), the drug is in the abdominal cavity. It is injected, but temperature consistency is not guaranteed.
When a temperature difference occurs (TD = 1℃), the LUT (U3 in Figure 5 (b)) sends a command signal and designs the coding so that the heater can heat or cool (Figure 6). And the designed coding guides the circuit to operate.
This design process is presented in Figures 4 and 5, and the designed circuit was manufactured using a PCB board and measured using a phantom and a thermal imaging camera sensor as shown in Figure 8. As a result of the measurement, it was possible to observe a phenomenon to maintain the treatment temperature (41-43℃) whenever the temperature changed (<41℃, >43℃).
- Question
Are there plans for further experimentation or clinical trials to validate the effectiveness of the proposed system?
- answers
Please refer to the lines of 492-515 (gray), lines of 515-520 (sky blue), and lines of 520-542 (gray) in the discussion.
As shown in Figure 5, Figure 6, and Figure 8, the performance effectiveness of the system was evaluated through design, simulation results, manufacturing, and measurement. In order to verify the validity of these performance evaluation results, animal experiments are performed and a thermal imaging camera is used. So, we observed changes in temperature and obtained results (Figure 9) through measurements to evaluate temperature changes and control performance effectiveness in the field.
In the future, we plan to conduct clinical trials through medical device licensing and pursue advancement of the system by investigating issues that need to be supplemented through clinical trials.
- Question
. What are the future directions for research or improvements in the proposed system?
- answers
Please refer to the lines of 415-515 (gray lined sentences), lines of 515-520 (sky blue), and lines of 520-542 (gray lined sentences) in the review.
- Question
Could you provide more details about the animal experiments mentioned for the future commercialization of the system? What parameters will be evaluated in these experiments?
- answers
The sentences in the lines of 515-520 (sky blue) have been supplemented. Please note.
- Question
Are there any specific technological aspects or components that differentiate this proposed system from existing temperature control methods in surgical treatments?
- answers
The specific story is lines of 39-65 (yellow), lines of 65-69 (red), lines of 70-77 (green), lines of 77-92 (sky blue), and lines of 93-97 (yellow) in the introduction. Please refer to the sentences in lines of 97-101 (black) and lines of 105-114 (sky blue).
See also lines of 405-463 (green) and expressions (9)-(11) in consideration. See also Figure 1, Figure 2, Figure 3, Figure 12, and Table 4.
- Question
How scalable is the proposed system, and what considerations have been made to ensure its adaptability to various clinical settings?
- answers
Completeness was improved through consideration and supplementation. Please refer to the lines of 501-514 (underlined gray), lines of 515-520 (sky blue), and lines of 520-542 (underlined gray) in the review.
Also, please refer to the lines of 543-546 (dark blue) in the last line of the review.
Here are some things to keep in mind:
Since the proposed system is a HIPEC surgical system, it can be used to treat colon cancer, ovarian cancer, and peritoneal cancer, so its utilization is expected to increase in the future through research and development, and demand for use in medical settings is also expected to increase [41, 42, 43].
- Question
What steps or plans are envisioned for the commercialization of this system, considering the need for animal experiments and subsequent regulatory approvals?
- answers
Completeness was improved through consideration and supplementation. Please refer to the lines of 501-515 (underlined in grey) in the discussion.
- Question
Are there any potential barriers or challenges anticipated in the commercialization process, and how do you plan to address them?
- answers
Completeness was improved through consideration and supplementation. Please refer to the lines of 501-515 (underlined in grey) in the discussion.
- Question
Can you provide insights into potential future research directions or improvements that could be made to enhance the proposed TC method?
- answers
Please refer to the lines of 364-372 (pink) in the discussion, Figure 11, lines of 364-372 (pink), equations (3)-(8), and lines of 385-402 (pink).
- Question
Conclusions should be more concrete, and future research directions presented.
- answers
The conclusion was expanded to be more specific and focus on future research directions.
- Question
Overall, the work is interesting; it just needs to follow the suggestions to improve the manuscript.
- answers
Thank you very much for reviewing my thesis and giving me a positive evaluation.
Round 2
Reviewer 2 Report
Comments and Suggestions for Authors
Reviewer' 1 comments:
Reviewer #:
After a thorough review of the author's revised manuscript (sensor-2800407), I conclude that they have provided his answers to the reviewers' suggested comments with appropriate explanations.
Therefore, I accept and recommend this version related to MDPI journal of sensors.